# The Effects of External Financial Support on the Capacities of Educational Nonprofit Organizations

**Oto Potluka [1],\* and Lenka Svecova [2]**

[1]   Center for Philanthropy Studies, University of Basel, Steinengraben 22, 4051 Basel, Switzerland
[2]   MIAS School of Business, Czech Technical University, Kolejní 2637/2a, 160 00 Prague, The Czech Republic
\*   Correspondence: oto.potluka@unibas.ch; Tel.: +41-61-207-2840

**Abstract:** Official development assistance provides an immense flow of financial funding to educational nonprofit organizations (NPOs). This source of funding faces criticism because of the unintended indirect effects it has in lowering the relative level of local NPOs' capacities. Our contribution addresses NPOs' financial capacities in an OECD country that receives a vast inflow of EU funding; namely the Czech Republic. To answer the research question on what impact the external financial assistance has on capacities in NPOs, we applied propensity score matching to a sample consisting of 633 educational NPOs covering the years 2006–2013. EU-funded NPOs report higher levels of real revenues, but not real assets, than non-funded NPOs. The EU funding helps in the short-term to improve NPOs' budgets, but not to increase assets.

**Keywords:** nonprofit organizations; financial capacities; assets; cohesion policy; the Czech Republic

## 1. Introduction

Official development assistance (ODA) and funding from foundations provide an immense flow of financial support to educational nonprofit organizations (NPOs) in developing countries. NPOs play an essential role in both donors and recipient groups. By delivering almost one-third of all the international financial help that is received by low-income countries, the private philanthropy sector plays a fundamental role in delivering development assistance [1], especially in education [2]. Moreover, NPOs form a crucial group of stakeholders that implement international assistance in the recipient countries.

Still, questions relating to the role of the local NPOs who implement international projects remain. How does international assistance influence these NPOs? Are they sufficiently well equipped to achieve their missions? Some studies comment that ODA damages the recipient country's local capacities. This problem relates to situations where local capacities are employed to manage foreign projects [3] and where nonindigenous knowledge has to be implemented [4]. Unintended effects of ODA assistance also arise in situations where the governments of recipient countries use their budgets to finance political goals that differ from those aimed at by international assistance. For example, a country whose government receives ODA assistance might consequently spend its resources on military objectives instead of expending them on the ODA objectives that are set out for low-income countries [5] or on the health care sector as set out for middle-income countries [6]. Although education seems to be less vulnerable to the crowding out effects [6] than other fields of ODA, still it is a component of international financial assistance within the framework of ODA.

The importance of education is underlined by the mean proportion of people aged 25–64 years attending further education and training courses in the most competitive and innovative economies in Europe, e.g., Switzerland 29%, Iceland 26%, and Norway 19% [7,8]. To implement educational programs beyond the school system, the providers need sufficient capacities to do that.

The state's educational system in the Czech Republic, our country of interest, is at a high level similar to the EU average. See, for example, the statistics on underachieving in mathematics, reading, and science [9]. Beyond compulsory education, the importance of continuing vocational education and training is growing, but still adult participation in learning and tertiary educational attainment are below the EU average [9]. Thus, when the obligatory system does not provide sufficient educational opportunities, informal educational channels can help and provide them to those who are willing to educate themselves.

Our contribution addresses similar issues. Knowledge about how local NPOs' build financial capacity would promote more effective implementation of international assistance and facilitate sustainable effects in education. Instead of focusing on low-income countries, we concentrate on an OECD country that is categorized as one of the world's emerging countries: the Czech Republic. It is ranked No. 40 in per-capita wealth on a list comprising 191 countries [10]. In global terms, it is a relatively wealthy country. Within the EU, it is one of the countries whose economic development is lagging and thus it is a major recipient of EU assistance provided by the EU Cohesion Policy. Not only is it a country with a vast inflow of external funding from the European Union, but it also belongs among the EU member states that received the highest per capita allocations from this funding during the period 2007–13. Thus, our research aims to shed light on whether an OECD country copes with similar issues when receiving substantial foreign assistance, as do the countries receiving ODA. We raise the research question of what impact does the external financial assistance have on capacities in NPOs.

The financial data on 633 Czech educational NPOs in the period 2007–13 enables us to examine how their financial capacity develops. We analyze whether local NPOs' capacities are improved by receiving exogenous funding for education and human resources programs that is financed by the EU. Thus, our analysis extends beyond understanding the extent of international assistance to low- and middle-income countries so that it can establish whether there are general principles at work in international financial assistance affecting the funding capacity of local NPOs.

The remainder of the study is organized as follows. In Section 2, we discuss the official development assistance of the educational sector and the philanthropic assistance that it receives from the perspective of coexisting fund providers: public, private, and nonprofit contributors. Moreover, we discuss how NPOs build financial capacity. Our study uses the framework of the EU Cohesion Policy funding for education and human resource development provided to the Czech Republic. The third section defines the data and applies the propensity score matching method. In Section 4, we discuss the results confirming the positive effect of EU funding on revenues but revealing no effect on the assets of local NPOs. The last section concludes with some implications for international assistance programs.

## 2. Official Development Assistance and Nonprofit Organizations in Education

### 2.1. The Coexistence of the Official Development Assistance and Private Philanthropy in Education

The public sector and NPOs could coexist and complement each other in order to increase the effectiveness of the provision of services. In the case of education, private philanthropic assistance is more targeted but less well coordinated than the ODA programs [11], and cooperation between these sectors is needed [12]. Steinberg [13] summarizes the problematics of multi-sectoral service provision in terms of a circular theory: The sector that assumes the leading role will vary depending on the nature of the situation, while the other sectors follow. Salamon [14–16] provides clarification of the simultaneous multi-sectoral provision of services in his 'voluntary failure theory'. He sees provision of public and NPOs' goods as complements, not as substitutes. Thus, the public sector and NPOs could coexist and complement each other in order to increase effectiveness of provision of services.

It is the interest of both developing and developed countries to make international development assistance effective and efficient. Nonprofit organizations should also contribute towards this, both on

the philanthropic donors' side and the beneficiary's side. The intermediary role of NPOs, therefore, relates primarily to situations where both markets and governments fail, and NPOs can fill this gap.

NPOs play a vital role in the implementation of international assistance programs. The financial funding enables NPOs to implement their missions and to achieve the goals of the donors. Their success is evidence of the achievement of their missions [17]. This is especially important for situations where NPOs implement projects that have systems of education that operate side by side with the governmental systems [12]. In the case of the Czech Republic, educational NPOs do not usually use strategic approaches [18], which is a weakness concerning the sustainability of their educational programs, as they tend to follow public grants to get funding for their activities.

In the Czech Republic, various sectors see the importance of education differently, or are differently capable to invest in it. General government expenditure on education measured as a share of GDP shrank to 4.5%, which is below the EU average [9]. On the other side, Czech firms are aware of the importance of education and skills for their competitiveness. Thus, they educate their employees; about 90.6% of firms send their employees to various training programs [9]. This still leaves some population behind. It concerns especially those who are unemployed. These people are a specific concern of NPOs who provide them with various training opportunities, including those financed by the EU funds.

### 2.2. Sustainability of Capacities when Implementing International Assistance

To implement ODA successfully, the receiving countries need to have sufficient capacities. Usually, countries which need the assistance the most are also the countries that have weak institutions, which undermines the effectiveness of the assistance programs [19]. The inefficiency of these countries' institutions and their lack of capacity to implement ODA programs effectively weaken the commitment of the stakeholders that wish to support them and lowers the amount of funding that will be disbursed by the ODA [19,20]. For example, donors have invested a great deal of effort in supporting the principle of general enrolment (education for all). This led to an increase in enrolment; however, it also led to a decrease in the quality of education provided [21], because the capacities of the available educational facilities became overstretched. Thus, building sustainable capacities is a challenging task [12] and there is a need to coordinate effort among objectives.

The ODA target countries became dependent on foreign aid. Heyneman and Lee [3] argue that this was the consequence of the lack of sufficiently sustainable capacity in these countries. Know-how imported from donor countries results in the target country's underutilization of its local resources [4], because it uses its capacities to implement projects that involve nonindigenous knowledge. Assistance of this kind causes path-dependence and limits efficiency [22]. Furthermore, it makes these regimes vulnerable to aid volatility [3,12].

The concept of sustainability concerning a country's development capacities can be seen from different points of view. It covers managerial and socio-cultural aspects [12,23], or political, financial, and institutional factors [24]. All of the studies reviewed by our research reflect financial sustainability as being the most crucial component of sustainability for the implementation of projects and programs as it does affect other types of capacities of NPOs.

### 2.3. Sustainability of Capacities of NPOs

To meet the requirements of their role, NPOs need to have sufficient capacities. Thus, the financial management of NPOs and measurement of NPOs' financial capacities became of high importance in the time of financial austerity and of high interest of researchers [25–32]). We rank financial capacity as being the most critical capacity, as it affects all the other types of capacity [23,24,27,33–35]. Sufficient funding enables NPOs to hire enough human resources [36,37], buy equipment and services [23], get knowledge and expert skills at an appropriate level, and to establish political networks [35] and social networks through social media [38].

Financial capacities help NPOs to fulfil their missions. Based on resource dependence theory, Lu, Lin, and Wang [39] point out that vulnerable NPOs have problems when reducing net assets, net earnings, or program expenditure; thus, these authors define financial capacities based on revenue growth, surplus margin, fixed assets growth, and expense growth. Achieving an optimal mix of revenues from various sources becomes a crucial task for NPOs' leaders. Financial independence with stable revenues enables NPOs to keep their mission [40]. From this perspective, Wilsker and Young [41] discuss the private good and public good nature of services and its influence on the revenue structure. They conclude that NPOs with private goods are more likely to earn their own revenues and to be more independent. In contrast, services that are characterized as public goods, such as basic education, are more reliant on public or charitable funding. This distinction is important, especially in times of financial austerity when other sources of funding are not available [26,28–32]. It also explains why NPOs with diversified revenues perform better [42]. Nevertheless, results of studies on concentration of funding on one source are inconclusive. There are arguments supporting concentration [42,43], but also opposing it, as NPOs can lose some freedom in implementing their missions and thus needing to cope with the financial pressure [44].

For our research, we concentrate on real assets and real revenues as proxy indicators for financial capacities in NPOs. In the long-term, the development of an NPO's real assets could be used as a proxy variable for its financial capacities [27]. The assets enable NPOs to generate revenue. Important for the NPOs is that the assets should grow in real terms to provide a sustainable basis for revenue. With these revenues, NPOs can fulfill their missions successfully.

*2.4. EU Funding to Education and Human Resources in the Czech Republic*

The EU Cohesion Policy accounts for approximately one-third of the EU budget. This policy includes the following objectives: the creation and sustainability of jobs, competitiveness, economic growth, sustainable development, and the quality of life [45]. The main tools that this policy uses are: (a) the European Regional Development Fund (ERDF); (b) the Cohesion Fund (CF) for investment in tangible assets; and (c) the European Social Fund (ESF) that is used to develop human resources.

During the years 2007–13, the Czech Republic implemented 26 programs funded by the EU Cohesion Policy. These represent an investment amounting to approximately EUR 16.7 billion, which was among the highest per capita investment made to an EU country in that period. In the Czech Republic, almost EUR 0.56 billion was invested in 1318 NPOs in 2824 projects until the end of 2013 [46,47].

The programs funded by the ESF have begun to attract the attention of NPOs in the Czech Republic. Almost 80% of Czech NPOs' applications to the EU for funding were for ESF assistance during the period 2007–13 [46]. Czech NPOs prefer smaller projects that are oriented towards social issues or education, as these are closest to their missions. Similarly, Cartier-Bresson [20] argues that NPOs are viewed as potential international assistance providers, but only for micro-projects that impact localities; i.e., they are not macro-projects that have a national impact.

The success rate of Czech NPOs in applying for EU Cohesion Policy funding is 31.7% [46]. This means that more than two-thirds of all NPOs' applications relate to capacities that are available to be employed elsewhere, given that they have not been engaged to implement the proposed projects. The high number of rejected project applications (more than 6000) imposes a significant burden on NPOs, as they have to finance the preparation of these applications, as the human and financial resources could be used for another purpose.

We use the case of the EU Cohesion Policy because it is similar to the ODA. In both cases, supranational funding bodies provide investment to local educational projects that have to be implemented according to the donors' objectives and principles.

## 3. Methods

This article draws on a quantitative study that investigates the relationship between exogenous financial support and the capacities of educational NPOs. Our analysis focuses on NPOs' financial

capacities (size of assets and revenues) as a precondition for their other activities. We applied a quasi-experimental approach by using propensity score matching to estimate the impact of EU funding on educational NPOs' financial capacities. In cases where there is a significant difference in the impact variable between the treated group and the control group in the pre-intervention period, this method is used in combination with the difference-in-differences method. Our initial sample consists of data for 633 educational NPOs in the Czech Republic and their official financial records for the period 2006–2013.

The Czech system defines NPOs based on an explicit list of legal forms. There are societies (former civic associations; 82,597 registered units + 24,739 branches of societies), public benefit corporations (2710), institutes (established for the public benefit purpose; 142 units), foundations (490), endowment funds (1331), and registered legal entities (established by religion organizations; 4117 units) [48]. Such a definition is also authoritative for public administration in relation to the nonprofit sector.

Educational NPOs that provide public-benefit activities accounted for approximately only 1% of all NPOs in the Czech Republic in the period 2006–2013. The majority of NPOs were membership associations—approximately 60% of them [48]. Since that time, the role of public-benefit corporations (PBCs) increased, especially after the reform in 2014, when the associations needed to be newly reregistered (Data on how many inactive associations did not reregister are still not available, but we can estimate that there were dozens of thousands of associations). The pubis belong currently with the registered associations to the growing legal forms of NPOs, which provide public benefit services (registered associations can provide services also solely to their members). The importance of educational NPOs is underlined by the fact that they produced approximately 9.3% of total NPO output and account for 11.7% of NPO employment in the surveyed period [48].

The educational NPOs in our analysis have the legal form of a public-benefit corporation (PBCs) (We use the term PBCs when we refer to the public-benefit corporations specifically. The term NPOs denotes nonprofit organizations generally (including PBCs)). Only 2.1% of all NPOs in the Czech Republic had this legal form in the surveyed period, but the advantage of this legal structure is that the NPOs have always been obliged to be transparent and to publish their annual financial records. Thus, we can collect data on their financial development and financial capacities. In the surveyed period (2006–2013), PBCs did not compose the major part of NPOs. In contrast to associations, they all had to be active, while the majority of associations was inactive and nonfunctioning, or simply did not report any activity. Thus, the PBCs provide us with a unique opportunity to study their financial records while the associations do not.

Our analysis focuses on educational PBCs. This group encompasses 23.3% of all PBCs in the Czech Republic. The fact that educational PBCs obtained more than 31% of the total EU funding allocated to Czech PBCs documents their privileged position. Moreover, they hold 32.8% of the overall assets held by Czech PBCs and 29.0% of the respective revenues [49].

*3.1. Data*

We used the following databases to compile a dataset for our analysis. The Database Albertina, which provides information about all registered organizations in the Czech Republic, the monitoring system of the ESF in the Czech Republic (Monit7+), the Central Register of Subsidies of the State Budget CEDR III, and the monitoring system of the Ministry of Regional Development of the Czech Republic (information about funding from the ERDF and CF). These databases enabled us to complete the dataset with the following variables:

- The Czech Statistical Office gives the unique **identification number** to all organizations.
- **Region** according to the registration address of each organization. We use this variable to indicate three main regions—Prague, the western part of the Czech Republic (Bohemia), and the eastern part of the Czech Republic (Moravia) (see Table 1).

- Classification of Economic Activities **CZ-COPNI**. This variable enabled us to select a subsample of educational PBCs from the whole population of these organizations.
- Year when a PBC was **established** and registered.
- Dummy variable to identify organizations supported by **EU funds during 2004–06**.
- Similar to the previous variable, but identifying organizations supported by **EU funds during the period 2007–13.**
- The amount of **EU Funds spent** by a particular organization in each year of the reference period (reported in CZK: we convert these values into EUR using the exchange rate of 27 CZK/EUR).
- The measure of **the total revenue** of a PBC's income during the period 2007–13 (reported in CZK; we converted these values into EUR using the exchange rate of 27 CZK/EUR). We drew our main financial data on revenues and assets from each of the PBC's annual reports and financial statements.
- **Total assets** presented on the balance sheets for the years 2007–13 (reported in CZK; we converted these values into EUR using the exchange rate of 27 CZK/EUR).

The total number of educational PBCs in the sample is 633. Of the 2721 PBCs registered in the Czech Republic, we had to exclude those who had specializations other than education, cases with errors in the reports, and cases without any reports. See Table 1 for information about the geographical distribution and size of the PBCs receiving EU financial assistance in our sample.

**Table 1.** Distribution of educational PBCs according to region of their registration and EU funding.

| | | Region | | | Support | | Total |
|---|---|---|---|---|---|---|---|
| | | **Prague** | **Bohemia** | **Moravia** | **No** | **Yes** | |
| Educational PBCs | N | 218 | 221 | 194 | 512 | 121 | 633 |
| | % | 34.4% | 34.9% | 30.6% | 80.9% | 19.1% | 100.0% |
| Educational PBCs per 100,000 inhabitants | | 17.2 | 4.3 | 4.8 | | | 6.1 |

Source: Own calculations based on Albertina, Monit7+, Commercial Register, and CEDR III.

For the analysis, we apply the total real revenues and total real assets (revenues and assets adjusted for inflation). It enables us to test whether the EU funding has an impact on financial capacities in educational PBCs and whether these PBCs are dependent on the EU funding. The year 2007 serves as a pre-intervention period in which none of the PBCs received the assessed financial assistance. The year 2013 serves as the post-intervention period in our sample.

*3.2. Propensity Score Matching*

Our approach to the analysis is based on propensity score matching. This quasi-experimental method compares two groups of cases [50,51]. The first group contains cases (PBCs in our study) that obtained treatment (EU financial assistance). A second group is a control group of PBCs that have not received any EU financial assistance. The only difference between these two groups is the financial assistance for the treated group, while the control group has not received any such assistance. For all the other characteristics, there must not be any statistical difference.

We used the variables describing each NPO in the pre-intervention period (i.e., region of registration); the year when an NPO was established; whether the NPO received funding during the program's period 2004–06; the size of assets in 2007, and the size of revenues in 2007. Researchers usually also use legal form as one variable characterizing an organization [52]. In our case, we use only one type of legal form, thus the PBCs have similar (if not the same) managerial structures across the whole sample based on the requirement of the law. We applied a logistic regression on these variables to estimate the propensity score, which represents the probability of an NPO obtaining EU financial assistance based on the above-mentioned variables. The approach of the nearest neighbor matching with replacement was used [53].

From the initial sample of 633 PBCs, we obtained a group of 136 funded PBCs and a control group of 72 PBCs. The limited size of the sample resulted from missing data for many PBCs in the year 2007. Both of these groups are statistically similar regarding the variables we used in the pre-intervention period. This means that there is no statistically significant difference between the compositions of these two groups in their average year of establishment, total revenues, total assets, and the place of residence of PBCs.

## 4. Results and Discussion

Our estimation reports a statistically significant difference in revenues in the post-intervention period. As there was no such difference in the pre-intervention period in the sample constructed by the propensity score matching, this reflects that we estimate a positive impact of the EU financial assistance on revenues in educational PBCs (see the Table 2). The real revenues in supported PBCs are five times higher than in the PBCs without support (0.79 million EUR vs. 0.16 million EUR) in the year 2013.

**Table 2.** Estimation of differences in revenues and assets.

| | Levene's Test for Equality of Variances | | *t*-Test for Equality of Means | | | N |
|---|---|---|---|---|---|---|
| | **F** | **Sig.** | **Sig. (2-tailed)** | **Mean Difference** | **Std. Error Difference** | |
| Real revenues (2007) | 0.114 | 0.736 | 0.419 | 192.3 | 237.1 | 57 supported 40 control group |
| Real revenues (2013) | 9.601 | 0.003 | 0.010 | 627.4 | 237.1 | 55 supported 41 control group |
| Real assets (2007) | 2.289 | 0.134 | 0.276 | 287.9 | 262.5 | 52 supported 39 control group |
| Real assets (2013) | 0.283 | 0.596 | 0.697 | 140.9 | 361.0 | 54 supported 39 control group |

Source: Own elaboration based on the data from www.justice.cz; the values of the mean difference are in thousands EUR.

In the data, there are a few outlying cases. These PBCs achieve much higher values of revenues and assets than the rest of the sample. We could use a combination of the propensity score matching with trimmed means [54], but it would further decrease our sample. Thus, in the following text, we comment mainly on the median values of tested variables. They represent the actual development better than average values. For example, a massive investment in assets in a few training centers caused the value of average real assets to skyrocket in the control group in the years 2012 and 2013 (see Figure 1).

From our analysis of the revenues and assets of the educational PBCs, we report four findings. First, the educational PBCs from the control group report lower real revenues than the PBCs receiving EU financial assistance over the whole surveyed period, except for the year 2007 (see Figure 1 and Table 2). Superior managerial skills may explain this in the funded PBCs and their orientation towards grant projects that include EU funding, and their fundraising activities [49]. This explanation is supported by the research of the Government Council for the nonprofit organizations which states that there were only approximately 100 professionalized NPOs in the Czech Republic in the year 2007 [55]. Moreover, Kovách and Kučerová [56] describe the creation of the so-called "Project Class" concerning the implementation of the EU Cohesion Policy in Central Europe. The emergence of this brand-new social class is strongly linked to the implementation of this policy, and its main activity involves the preparation and implementation of the projects funded by the EU funds, as well as providing associated consultancy services. The supported PBCs belong to this class. Thus, they would have higher revenues than the PBC's in the control group even without EU funding, as their business strategy

draws opportunistically on available resources. Unfortunately, information about managerial skills is not a part of the dataset; otherwise, we would use it in the propensity score matching estimation.

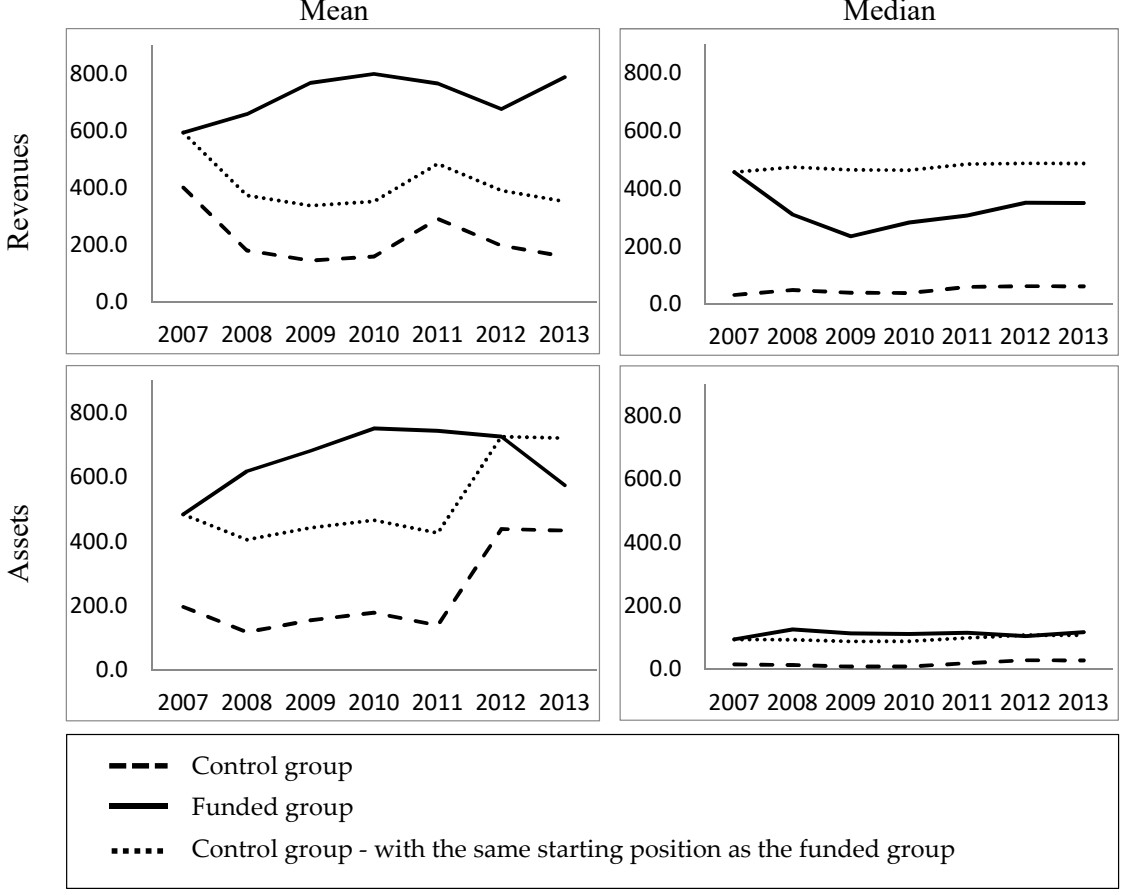

**Figure 1.** Development of real revenues and assets in both groups of educational PBCs. Source: Own elaboration based on the data from justice.cz; All values are in EUR thousands.

Second, our analysis revealed the dependency of the educational PBCs on the current economic situation in the country. The economic downturn in the years 2009, 2012, and 2013 hit the educational PBCs because the donors from both the private and the public sectors were cautious and shrunk their financial support to NPOs [57]. These cutbacks were especially evident in the revenues of the NPOs in the year 2009. The austerity was also reflected in PBCs' decreasing real revenues, as in times of economic growth they witnessed growth. This finding is obvious and can be expected, but it reveals that PBCs are weak in multisource fundraising and depend on one main source of funding. From this perspective, public budgets constitute the main source of funding and are responsible for approximately 50% of all revenues earned by funded PBCs, but only approximately 30% of all revenues earned by the control group of PBCs. Still, the percentage is lower than that for the NPO sector as a whole, where it is reported to be approximately 60% in the Czech Republic [58,59]. Concentration on one core funding resource need not necessarily be a problem, because, as Schnurbein and Fritz [43] show, such organizations grow; however, a PBC's sole dependency on public grants can be problematic if its future is not clear, and another source of subsidy cannot readily restore the loss of that funding. Generally, this resource dependence poses a problem for NPOs [39].

Third, median real revenues also reveal the reliance of some NPOs on EU funding. In the years 2007 and 2008, there was a minimal offer of new calls for proposals based on the allocations assigned for the programs 2004–06. Moreover, there was a delay in implementing the new programs that were released in the period 2007–13. Thus, the decreasing trend reflects this situation. The turning

point occurred in the year 2010 with new calls for proposals and freshly implemented projects. These findings confirm how significant EU funding is for the revenues of PBCs that can obtain it. On the other hand, it confirms path dependence [22,49]. It also confirms the resource dependency theory [39] accompanied by the tendency of PBCs to implement supranational projects instead realizing their missions. The share of EU funding in PBCs' revenues grew from 9.8% in 2006 (financed by the programming period 2004–06) to 42.3% in the year 2013.

The funded PBCs achieved positive net revenue throughout the surveyed period. In the years 2007, 2008, 2011, and 2012, they achieved this only because of the EU funding; without it, they would have been in deficit. The PBCs in the control group achieved small but positive surpluses throughout the surveyed period. This also underlines the financial vulnerability of these PBCs. It is similar to the plight of organizations that implement international assistance programs in low-income countries [3,12] and the Czech NPOs that do not use a long-term strategic approach [18]. This situation is caused by "cheap money" from the EU Cohesion Policy. There is no requirement for ESF assistance to be co-financed. In the case of the ERDF, assistance must be co-financed by approximately 50%. Thus, it is clear that more than 80% of all EU Cohesion Policy applications submitted by Czech NPOs relate to ESF assistance. The median revenues of the control group remained stable during the years 2007–2013.

Fourth, the development of real assets is more or less stable in both surveyed groups of PBCs. Thus, the estimators of the impacts of the EU funding on real assets are statistically insignificant for the year 2013. The increase in average real assets in the control group is caused by the above-mentioned educational centers in which investment in assets appeared. Both groups reported a more or less stable development of median assets throughout the surveyed period with a slow decline during the economic crisis. On the other hand, the increase also indicates that neither funded PBCs, nor PBCs in the control group were able to increase their level of real assets over the long-term systematically. In compliance with the study of Lu, Lin, and Wang [39], without stable or growing real assets, NPOs have problems as it can cause lower income and lower program expenditures. Thus, their long-term viability is dependent on their current sources of funding, as they do not generate higher revenues from the increasing value of their current assets [27] and they do not invest in assets. It presents a very similar situation to ODA where capacities are devoted to managing foreign projects [3] with balanced budgets. Financial reserves may not be built or invested in assets. Moreover, such projects usually implement public policies [49] that NPOs are unable to influence [60] and make no use of indigenous knowledge [4].

From the perspective of public budgets, the financial effect also occurs with EU funding. Although this effect is insubstantial in the case of education [6], it is nevertheless present in ESF projects, where salaries form a significant part of expenses. The EU covers 85% of the ESF-funded projects, while the national authorities contribute only 15%. Depending on the size of the salary, up to 50% of the gross amount is deducted for income tax, and social and health contributions. Thus, the public budget's tax and insurance revenues are higher than the contributions paid towards it (even when counting contributions of the Czech Republic to the EU budget).

These findings confirm that international assistance should not only contribute benefits from the donors' knowledge, but it should also enable local NPOs to build their capacities. For example, the EU Cohesion Policy's programs include a program that is dedicated to technical assistance aimed at increasing expert capacities and expanding general absorption capacity (to use the EU funding wisely with an impact). This type of assistance is usually beyond the reach of ordinary NPOs. NPOs usually only implement short-term projects with balanced budgets with a limited possibility to invest in long-term assets. Thus, it would be advantageous to enable NPOs to use part of the assistance to develop their institutional capacity, even if it would be non-financial assistance, such as consultancy or training. Long-term increase in human and institutional capacities will also help NPOs to work more efficiently and to build their financial capacities in the long-term. In the Czech Republic, the assistance of this type, which was managed by the Civil Society Development Foundation, was only made available for a short period during 2004–06 [61]. Although this policy only existed for a brief time,

it has been positively evaluated by NPOs [61]; recently, new attempts have been made to support NPOs' capacities.

The participation of capable local stakeholders helps to improve the relevance of the programs and promotes reaching a consensus [62]. While it is generally accepted that the public sector is a partner for international development assistance, albeit with minimal possibilities to influence decisions [63], the involvement of NPOs is still limited [64]. Thus, the capacities of local stakeholders are important for setting up efficient international programs.

## 5. Conclusions

Our research confirmed that the situation of recipient NPOs in an OECD country is very similar to the situation of NPOs in low-income countries. Although we conducted our research on one of the richer countries in the world, we found that the effects of EU financial assistance on NPOs were very similar to the effects that ODA has on NPOs in low-income countries. The effects of assistance on NPOs relates to the dependence of some NPOs on external international funding and the endangered financial viability of the projects where there is volatility in the provision of international financial assistance. Furthermore, these weaknesses lead to the implementation of supranational policy goals and the NPOs' neglect of their missions, as they have to decide either to apply for funding and implement the international programs' goals or to implement their missions, most often without any external assistance.

Our estimates of the development of the PBCs' long-term financial viability [27] do not substantiate either negative or positive effects EU Cohesion Policy. Our results have not delivered proof of any significant difference between funded and non-funded PBCs in long-term financial viability measured as real assets. Thus, we cannot say that international financial assistance has harmed the financial capacities of the Czech educational PBCs, nor can we confirm that it has helped to increase them. The positive estimates of the impact of the assistance to revenues reveal that in the short-term, the assistance helped sustain activities in PBCs. Thus, we can conclude that the same behavioral principles apply to NPOs in rich countries and low-income countries.

Private philanthropy has already played an essential role in international development assistance. Although it does not provide sufficient resources to cover all the public funding gaps in ODA [3], private philanthropy can fill the gaps caused by market and government failures. Private philanthropy can target unmet local needs more precisely than ODA. Moreover, if the local NPOs have sufficient capacities, they could increase the efficiency of international assistance by providing information to local stakeholders and communicating them to international organizations.

To confirm or reject our conclusions, future analysis should be conducted on a larger sample of NPOs and on another OECD country, which receives large amounts of external funding. Moreover, a new initiative of the Czech Ministry of Labor and Social Affairs aimed at NPOs' capacity building enabled NPOs to obtain funding to increase their managerial capacities. When these projects are implemented at the end of 2018, they will provide us with an ideal opportunity to test our conclusions on another sample of NPOs and targeted intervention.

**Author Contributions:** O.P. and L.S. designed the research, analyzed the data, contributed to the discussion of findings, and wrote the final version of manuscript. O.P. has applied propensity score matching on the IBM SPSS software version 24.

**Funding:** This research received no external funding.

**Conflicts of Interest:** The authors declare no conflict of interest.

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
