# Peer review of "The Effects of External Financial Support on the Capacities of Educational Nonprofit Organizations"

_sustainability, doi:10.3390/su11174593_

Round 1

Reviewer 1 Report

In the paper "The effects of external financial support on the capacities in educational nonprofit organizations" while sharing the purpose of the research, this work can be considered, in its current form, more suited to a divulgative publication rather than appear in a scientific series.

At present the margins for an immediate and coherent revision of the article are not found. 

The paper should use a solid scientific methodology coherently motivating the results, statistically valid in order to give a suitable scientific value to the contribution.

Author Response

In the paper "The effects of external financial support on the capacities in educational nonprofit organizations" while sharing the purpose of the research, this work can be considered, in its current form, more suited to a divulgative publication rather than appear in a scientific series.

At present the margins for an immediate and coherent revision of the article are not found. 

The paper should use a solid scientific methodology coherently motivating the results, statistically valid in order to give a suitable scientific value to the contribution.

Response of the authors: We have improved the manuscript according to the comments of the other reviewers. We believe that responding to their comments makes the manuscript also acceptable for this reviewer.

Reviewer 2 Report

The study has discussed an interesting topic in relation to educational NPOs in Czech Republic. The author may consider some comments as following.

Introduction

The research questions and research aim should be clearly stated at the introduction section.

Line 56, the author used NGO as an alternative of NPO, it would be better to have a short explanation (maybe, in footnote) the similarity/difference between NPO and NGO in Czech Republic.

The author has discussed about the funding of educational NPOs in Czech Republic, however, the author should extend the discussion to cover the differences between stated/governmental education and educational NPOs.

Literature review

There is very little discussion of previous studies in this study, the author should consider adding further discussion of prior studies related to the topic.

Method

Data: the study is conducted based on a sample of educational NPOs. However, it seems that this sector is not significant in comparison to the whole NPOs in Czech Republic since the paper mentioned that educational NPOs accounted for only 1% of all NPOs in Czech Republic. This type of organization also produced less than 10% of total NPO output. In order to demonstrate the contribution of the study, author may consider other factors to emphasize the importance of studying this sector (educational NPOs).  

The measurement of educational NPOs’ capabilities in terms of revenue and total assets should be explained further. Author may also consider adding more information of different types of revenue generated by the Educational NPOs. If funding from EU is a part of NPO’s revenue source, the finding that funded NPO group has higher revenue than control group seems not so surprised.   

Results and discussion

The study concluded that “The EU funding only has a short-term effect, as it causes the NPOs’ revenues to become dependent on it”. I am not convinced with this conclusion since the comparison of assets and revenue of two group of NPOs may be not sufficient to assess the effect of EU funding.  

Others:

The author mentioned that educational NPOs from their study are public-benefit corporations (PBC). However, there is little discussion of how this type of organizations are different from other NPOs. The author should consider adding more discussion about this form of organization.

Author Response

The study has discussed an interesting topic in relation to educational NPOs in Czech Republic. The author may consider some comments as following.

Introduction

The research questions and research aim should be clearly stated at the introduction section.

Response of the authors: We have added the aim and the research question to be clearly stated in the introduction.

Line 56, the author used NGO as an alternative of NPO, it would be better to have a short explanation (maybe, in footnote) the similarity/difference between NPO and NGO in Czech Republic.

Response of the authors: The term NGO was used mistakenly in the manuscript. Now, we use only the term NPO thoroughly the text.

The author has discussed about the funding of educational NPOs in Czech Republic, however, the author should extend the discussion to cover the differences between stated/governmental education and educational NPOs.

Response of the authors: Thank you for the relevant comment. We have added a paragraph on the results of the obligatory educational system and why can the informal systems help (for example the education provided by NPOs).

Literature review

There is very little discussion of previous studies in this study, the author should consider adding further discussion of prior studies related to the topic.

Response of the authors: We have deepened the discussion on the role of NPOs by adding information and discussion of various literature on the theme of education, and the role of NPOs in it.

Method

Data: the study is conducted based on a sample of educational NPOs. However, it seems that this sector is not significant in comparison to the whole NPOs in Czech Republic since the paper mentioned that educational NPOs accounted for only 1% of all NPOs in Czech Republic. This type of organization also produced less than 10% of total NPO output. In order to demonstrate the contribution of the study, author may consider other factors to emphasize the importance of studying this sector (educational NPOs).

Response of the authors: The added value of the PBCs in our sample is that currently they belong to the mostly used legal form. In the surveyed period (2006-2013), PBCs did not compose the major part of NPOs. In contrast to associations, they all had to be active, while majority of associations was inactive and nonfunctioning.  We have used that period because the newer data set is still not available (for the programming period 2014-2020).

The measurement of educational NPOs’ capabilities in terms of revenue and total assets should be explained further. Author may also consider adding more information of different types of revenue generated by the Educational NPOs. If funding from EU is a part of NPO’s revenue source, the finding that funded NPO group has higher revenue than control group seems not so surprised.

Response of the authors: We have improved the part of the manuscript explaining importance of these two variables for fulfilling of NPOs’ missions. We have also added discussion on resource dependence.

Concerning the results, we have applied propensity score matching in a combination with difference – in – difference approach, thus even with initially different size of revenues or assets, the method can cope with it. We have added a line describing development of revenues and assets in the control group if they were the same in pre-intervention period to make it easier to compare development of these two groups.

Results and discussion

The study concluded that “The EU funding only has a short-term effect, as it causes the NPOs’ revenues to become dependent on it”. I am not convinced with this conclusion since the comparison of assets and revenue of two group of NPOs may be not sufficient to assess the effect of EU funding.  

Response of the authors: When applying the propensity score matching, the two groups of NPOs were statistically similar in pre-intervention period, including total revenues and total assets. The only difference available in our dataset was the difference in EU funding between the treated and the control group. Based on it, we can estimate the effects of EU funding. It was visible on the development of median of revenues during the period when there were no call for proposals (2007-2009). During this period, median revenue of treated group was declining.

We have corrected the text according to exact results of our analysis.

Others:

The author mentioned that educational NPOs from their study are public-benefit corporations (PBC). However, there is little discussion of how this type of organizations are different from other NPOs. The author should consider adding more discussion about this form of organization.

Response of the authors: After the reform in 2014, the PBCs belong currently to the most used legal form of an NPO. We have discussed this issue more to the detail in the updated version of the manuscript.

Reviewer 3 Report

In table 1 are empty spaces. It should be changed.

Figure 1 is not unintelligible.

Author Response

In table 1 are empty spaces. It should be changed.

Response of the authors: Thank you for the comment. We have added the average number of educational NPOs for the whole country. (Notice: Changes of the numbers for two regions are given by rounding). The empty cells in the “Support” columns are empty as it does not make sense to count the average number of NPOs in a region as there is no region defining it.

Figure 1 is not unintelligible.

Response of the authors: Thank you. Another reviewer asked for more explanation of the development, thus we have added a line describing development of revenues and assets in the control group if they were the same in pre-intervention period to make it easier to compare development of these two groups.

Reviewer 4 Report

This paper could make an important contribution, it contains a significant and interesting material and should be provided with the opportunity to be improve.

1 - The most important criticism here is on the second section, the background doesn't discuss a theoretical framework. In my view, the literature review lacks some important theoretical support. Thus, for example, the Resource Dependence Theory could explain the different effects on the capacities of NPOs. The Institutional Theory is also an important approach to explain the behavior of the indirect effect of long-term on the nonprofits organization’s capacities.

2 -The paper has an important empirical research, but, in my opinion, have lacks on the theoretical support, essentially in the educational and financial area.

3 - The research design section lacks an exhaustive description of what Czech Republic nonprofits are, mainly in their financial law and the context in which they operate, and I believe that a reference to the institutional theory can help in better explaining these aspects.

4 - In line 322-325 the authors mention “These findings confirm that international assistance should not only contribute benefits from the donors’ knowledge, but it should also enable local NPOs to build their capacities. For example, the EU Cohesion Policy's programs include a program that is dedicated to technical assistance aimed at increasing expert capacities and expanding absorption capacity.”is not completely clear if the authors refers to a financial capacity or to another type of capacity on the NPO. It would be interesting to have, in a deeper way, some extra information about.

5 - Maybe the authors can add some recent papers, since do not have any reference from 2018 or 2019.

6 - In the discussion the paper needs to linked the theoretical support with the results in a consistent manner in order to connected the conclusion section with the theoretical framework.

I had realy appreciated the opportunity to read this interesting contribution. I would encourage the authors to receive these comments as a constructive way to improve and strengthen their paper.

Author Response

This paper could make an important contribution, it contains a significant and interesting material and should be provided with the opportunity to be improve.

1 - The most important criticism here is on the second section, the background doesn't discuss a theoretical framework. In my view, the literature review lacks some important theoretical support. Thus, for example, the Resource Dependence Theory could explain the different effects on the capacities of NPOs. The Institutional Theory is also an important approach to explain the behavior of the indirect effect of long-term on the nonprofits organization’s capacities.

Response of the authors: We have added discussion concerning financial capacities in NPOs based on resource dependence theory. Moreover, we have added a discussion concerning institutional differences in NPOs in the data section to underline that we used one legal type of NPOs to have similar (if not the same) managerial structures across the whole sample.

2 -The paper has an important empirical research, but, in my opinion, have lacks on the theoretical support, essentially in the educational and financial area.

Response of the authors: We have added discussion on financial capacities in NPOs and information about the role of informal education.

3 - The research design section lacks an exhaustive description of what Czech Republic nonprofits are, mainly in their financial law and the context in which they operate, and I believe that a reference to the institutional theory can help in better explaining these aspects.

Response of the authors: We have added a paragraph concerning the Czech nonprofit sector together with a short discussion on legal form we use.

4 - In line 322-325 the authors mention “These findings confirm that international assistance should not only contribute benefits from the donors’ knowledge, but it should also enable local NPOs to build their capacities. For example, the EU Cohesion Policy's programs include a program that is dedicated to technical assistance aimed at increasing expert capacities and expanding absorption capacity.”is not completely clear if the authors refers to a financial capacity or to another type of capacity on the NPO. It would be interesting to have, in a deeper way, some extra information about.

Response of the authors: To make the paragraph more clear, we have narrowed the sentences and added information that it can be consultancy, or trainings, which could lead to increased financial capacities.

5 - Maybe the authors can add some recent papers, since do not have any reference from 2018 or 2019.

Response of the authors: We have added some recent literature.

6 - In the discussion the paper needs to linked the theoretical support with the results in a consistent manner in order to connected the conclusion section with the theoretical framework.

I had realy appreciated the opportunity to read this interesting contribution. I would encourage the authors to receive these comments as a constructive way to improve and strengthen their paper.

Response of the authors: We have added some parts to link our results and discussion with the theoretical discussion on resource dependency theory.

Round 2

Reviewer 1 Report

The paper has been significantly improved and the previously highlighted critical issues have been accepted and implemented.

Therefore the paper can be published.

Reviewer 2 Report

Thank you for revising the paper and response to reviewer's comments. I am satisfied with authors' response.